# Modeling Fate and Transport of Nutrients and Heavy Metals in the Waters of a Tropical Mexican Lake to Predict Pollution Scenarios

Jorge I. Alvarez-Bobadilla [1], Jorge O. Murillo-Delgado [1], Jessica Badillo-Camacho [1], Icela D. Barcelo-Quintal [2], Pedro F. Zárate-del Valle [1], Eire Reynaga-Delgado [3] and Sergio Gomez-Salazar [4,*]

1  Departamento de Química, Universidad de Guadalajara-CUCEI, Blvd. Marcelino García Barragán # 1421, Esq. Calzada Olímpica, Guadalajara 44430, Jalisco, Mexico; iq_alvarez@hotmail.com (J.I.A.-B.); george23877@hotmail.com (J.O.M.-D.); jessica.bcamacho@academicos.udg.mx (J.B.-C.); pedrozarate@hotmail.com (P.F.Z.-d.V.)
2  Departamento de Ciencias Básicas, Universidad Autónoma Metropolitana-Azcapotzalco, Av. San Pablo Xalpa180, Col. Reynosa Tamaulipas, Azcapotzalco, Mexico City 02200, Mexico; ibarceloq@gmail.com
3  Departamento de Farmacobiología, Universidad de Guadalajara-CUCEI, Blvd. Marcelino García Barragán # 1421, Esq. Calzada Olímpica, Guadalajara 44430, Jalisco, Mexico; eire.rdelgado@academicos.udg.mx
4  Departamento de Ingeniería Química, Universidad de Guadalajara-CUCEI, Blvd. Marcelino García Barragán # 1421, Esq. Calzada Olímpica, Guadalajara 44430, Jalisco, Mexico
*  Correspondence: sergio.gomez@cucei.udg.mx

**Abstract:** The tropical lake Chapala is an important source of drinking water in western Mexico since it supplies ~65% of the water consumed in the urban city of Guadalajara. To obtain different pollution scenarios, the presence of pollutants in this waterbody was modeled using a coupled hydraulic and transport model. Two water sampling campaigns were modeled. The governing equations were applied using the routines RMA2 and RMA4 in the Surface-Water Modeling System (SMS) software V 8.1. Hydraulic and transport models were calibrated to describe the water level, velocity, and fate of pollutants. The numerical model showed satisfactory results for the simulated data, analyzed against water level, current velocity, and pollutants measurement data through the Relative Percentage Deviation (RPD), except for ~20% of the sites and the 12-month simulation periods. The hydraulic calibrations showed that the dispersion coefficients were higher for nutrients compared to metals, indicating that the nutrients are dispersed throughout the lake and have a stronger impact on the lake's water quality. The hydraulic model simulations indicated the presence of points in the central-eastern zone, the lowest concentration of $PO_4^{3-}$, which can be attributed to the presence of vortexing. The metal simulations indicated that the dissolved Ni was the best approximation to the measured values. This is the first study on Lake Chapala regarding the modeling fate and transport of pollutants in relation to the prediction of pollution scenarios.

**Keywords:** Lake Chapala; RAM2; RAM4; SMS; nutrients; heavy metals

## 1. Introduction

The use of water for human consumption in large urban centers is a great concern worldwide due to shortages and the low availability of this resource. In this regard, the use of natural water bodies for water supply is the most widely used alternative, especially in developing countries [1,2]. However, uncontrolled industrial development and the formation of large human settlements have created pollution problems in these water bodies since substantial amounts of both municipal and industrial waste are dumped into them without any previous treatment. As a result, there is a need to develop tools such as prediction models of the fate of pollutants that can help obtain better control over the

water quality of natural water bodies that act as water supplies and can further contribute to efforts to maintain an equilibrium in aqueous ecosystems.

Simulation modeling is essential for predicting the environmental fate and potential environmental consequences of chemical pollutants. However, very few simulation models have been applied to tropical lakes [3–5]. With the application of coupled hydraulic and transport models of pollutants, it is possible to simulate their concentrations and independently predict important changes in water quality [6,7]. Therefore, transport processes must be included in these models to obtain better pollution scenarios. In this regard, Hansen and Maya [8] simulated the effects of variations in the water composition of Lake Chapala on the sorption of Cd and Pb in sediments to assess the migration behavior of these metals. Although knowledge of the migration behavior of metal pollutants in a lake is important for assessing the health status of the lake, their study's simulations were based on only three sampling sites, thus their models were non-representative of both point and non-point sources, compared to the present study, where 12 sites were included to strengthen the base line of the data to generate the simulations. Furthermore, computer-based modeling applied to solving environmental problems, such as the water quality of lakes, aids us in understanding the dynamics of contaminants in sediments and provides the information required to plan corrective actions for contamination control [9]. For example, Ajiwibowo [10] used the SMS software to study Kerinci Lake in the Province of Jambi, Sumatra Island, Indonesia, and formulate subsequent actions designed to protect the area. The model was then developed into a model of sedimentation and water quality, resulting in yearly changes in the investigated parameters. However, his work could be strengthened if he considered the effects of disturbances on the water velocities, such as vortexing, which can be present in the ecosystem with regards to the distribution of pollutants. Jennings [11] employed the package SMS to study the sedimentation and scour in the small urban Lower Shaker Lake in the Doan Brook Watershed of Cleveland, Ohio. The modeling perspective used in this work was demonstrated to be of great use in uncovering the systems at work in Lower Shaker Lake and was proposed for the evaluation of remediation strategies for other small urban lakes. Again, there is a gap between his study and the actual models in that the presence of further pollutants in this lake was not accounted for, and more comprehensive remediation strategies could be implemented. On the other hand, Hillman et al. [12] studied the bi-dimensional hydrodynamics of the Los Patos Lagoon in the state of Río Grande do Sul, Brazil, using the SMS software (RMA2 model). The simulation results permitted the authors to highlight the sensitivity of the ecosystem in this estuary (the Saco estuary) to wind action and level conditions by rendering these variables dominant over the entry and exit of water in the Saco de Mangueira. Koue et al. [13] conducted studies on numerical models applied to lake ecosystems, focusing on the thermal stratification of Biwa Lake, Japan, using a three-dimensional hydrodynamic model. The results of this work indicated that the seasonal change in thermal stratification was satisfactorily reproduced by the hydrodynamic model simulations. Based on these investigations, we selected the coupled model (hydraulic and transport) in the present work due to the fact that it can satisfactorily predict pollutant simulations of a 3D ecosystem by considering it as a bidimensional ecosystem.

The concentrations of toxic metals and nutrients can increase unexpectedly in Lake Chapala (which is a dynamic ecosystem) due to natural processes such as evaporation, wind action on the lake's surface, and the low flow rates of the Lerma River. Heavy metals such as cadmium, nickel, zinc, lead, and mercury are unnecessary for human health, yet they have high levels of toxicity at low concentrations and can bioaccumulate in body tissues over long time periods [14]. On the other hand, the nutrients can have negative impacts on the health of aquatic ecosystems, including harmful algal blooms, a decrease in biodiversity, unpleasant smells, and contamination of drinking waters, which can open the door for serious public health threats [15]. Considering these challenging issues, it is important for both public and ecosystem health that we acknowledge the transport and fate of toxic substances present in water bodies through modeling. Furthermore, in

the context of human rights to water [1], predictive models of surface waters used for human consumption by large cities can contribute comparative data together with water quality indicators to the 17 SDG volunteers, particularly in regard to goal 6.3, referring to indicator 6.3.2.

As a contribution to evaluations of the environmental health status of Lake Chapala, this study aimed to assess the impacts of physicochemical water quality variables such as nutrients (e.g., $NO_3^-$, $PO_4^{3-}$) and heavy metals (Mn, Cr, Cd, Cu, Pb, Fe, Ni, and Zn) on the pollution scenarios of Lake Chapala through the application of a coupled model of these substances. This is the first comprehensive study on the modeling of pollutants that can help predict possible pollution scenarios, which, in turn, will assist in the assessment of their roles in shaping the water quality of this lake as a starting point for future studies.

## 2. Materials and Methods

### 2.1. Study Area

Lake Chapala is in western-central Mexico, and its average depth is 4–7 m (Figure 1). It is the largest body of fresh water in Mexico and is used for domestic, agricultural, and industrial purposes and to support small-scale but productive fishery activity [16]. Lake Chapala is a shallow tropical water body. It is the largest and most important freshwater body in Mexico and supplies water for human consumption to the nearby city of Guadalajara, with approximately 5 million inhabitants [17], accounting for approximately 65% of the water consumed in this city [18]. The lake is about 1515 m above sea level, has a surface area of 900 km², and a maximum depth of 7 m [19] (Figure 1). In addition to being a water supply source for human consumption, the lake is used for fishing, recreation, and irrigation, and is a receptacle for industrial and municipal discharges. Nonetheless, increased levels of pollution caused by nutrients and heavy metals, which mainly derive from the Lerma-Chapala Basin, have prompted concern since Lake Chapala is their landing place (Figure 1) [14,20]. The main tributary is the Lerma River at the site of Maltaraña, and its natural drain is the Santiago River. There is an artificial drain at site S6 (Figure 1). This site is the water pumping station for the city of Guadalajara. Human activities around this lake during the last 50 years have modified this ecosystem. The main contributors to these changes have been inputs from the Lerma River, which carries discharges from the communities and municipalities located along the river downstream. Floating polluted particles from urban and industrial sites and the entire watershed are dragged into the lake by this river [14]. Potentially toxic pollutants incorporated in municipal, industrial, and agricultural discharges from locations along the entire watershed have Lake Chapala as their destination.

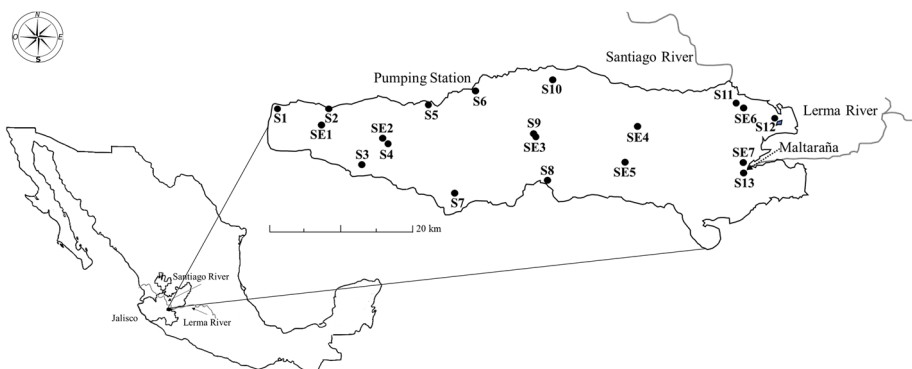

**Figure 1.** Location of Lake Chapala, showing both sampling and hydraulic measurement sites.

### 2.2. Water Sampling

Thirteen sites were selected for the collection of water samples during the dry seasons of May 2014 and May 2015. Sites SE1–SE7 were added to the 2015 sampling campaign in order to measure the water velocities. The selection of these sites was based on preliminary

simulations of the hydraulic model, which showed the existence of vortex formations at these sites originating from wind action and small currents beneath the water surface. The GPS site locations are shown in Table S1 (Supporting Information). Water sampling was conducted following the procedures established in the Mexican Standard [21] and the APHA manual [22]. Water samples were collected in 1 L polyethylene bottles previously soaked in 10% V/V $HNO_3$ for 24 h, rinsed with deionized water, and stored at 4 °C until further analysis.

## 2.3. Nutrient Analysis

The analysis procedures of the measured, $NO_3^-$ and $PO_4^{3-}$ concentrations were performed following the methods 4500-$NO_3^-$ A: nitrogen (nitrate) and 4500-phosphorous, respectively, established in the APHA manual [22]. All glassware was thoroughly washed with 15% V/V $HNO_3$ for 3–8 h, followed by washings with phosphate-free soap for 2 h. Finally, all glassware was rinsed with distilled and deionized waters.

## 2.4. Metal Analysis

*Total metals*. These are defined as the concentration of metals determined on an unfiltered sample after vigorous digestion [22]. Concentrated $HNO_3$ was added to the water samples from the field to lower the pH to <2. A digestion procedure was followed according to USEPA Method 3051 [23] in a microwave oven, CEM MARS-5 (Matthews, NC, USA). The resulting extracts with heavy metals were brought to the mark in a flask and analyzed by atomic absorption spectrometry (AAS) with a ContrAA 300 Analityk Jena instrument (Jena, Germany). The metals analyzed included those of environmental interest, namely Mn, Cr, Cd, Cu, Pb, Fe, Ni, and Zn. The detection limits of the instrument were as follows (mg $L^{-1}$): (a) Cd = 0.002, (b) Pb = 0.05, (c) Cr = 0.02, (d) Mn = 0.01, (e) Zn = 0.005, (f) Ni = 0.02, (g) Cu = 0.01, and (h) Fe = 0.02. The effects of background interferences were determined by running solutions containing all the contaminants except lake water. Deionized water with a resistivity of 18 MΩ (Barnstead, Chicago, IL, USA) was used for all the dilutions.

*Dissolved metals*. Are defined in Method 3030 B of the APHA manual as those metals of an unacidified sample that can pass through a 0.45 μm membrane filter [22]. Unacidified lake water samples were filtered through a 0.45 μm membrane in the laboratory. The samples were stored at ~4 °C until further analysis by AAS. For the particulate metal simulations, *particulate metal* concentrations were obtained by subtracting the total metal from the dissolved metal concentrations, according to the procedures in the APHA manual [22].

## 2.5. Measurement of Hydraulic Parameters

With the aim of validating the hydraulic model required to solve the mass transport model, hydraulic measurements were performed in the field at the selected sites around the lake. The following hydraulic parameters were measured: (a) the depth, measured using a probe of the Bottom-Line Side Finder brand Buddy 1200 model; and (b) the velocity and direction of superficial water, measured using a Vernier brand flowmeter with an interval of 0–4 m $s^{-1}$ and a resolution of 0.0012 m $s^{-1}$. The flowmeter was connected to an interface that stored the obtained data in 100 s intervals, providing the superficial velocity of the water. The water directions were determined with a composite pocket transit theodolite, Brunton ComPro brand, Quadrant model (Riverton, WY, USA), with a magnetic declination of 9° E.

## 2.6. Fate and Transport Models of Pollutants

The model used here is bi-dimensional and considers mass changes caused by advective and dispersive transport and by both chemical and biological reactions. The equations of the mass balances are solved using the routines RMA2 and RMA4 contained in the Surface-Water Modeling System (SMS) software. In both models, hydraulic parameters are adjusted according to the field-collected data. To verify the precision of the models,

the calculated concentrations of pollutants are compared with the measured concentrations. In this study, two sampling campaigns were simulated during the dry seasons of 2014 and 2015. The precision of the predictions was determined by considering the Relative Percentage Deviation (RPD) between the measured and simulated data and setting an error maximum tolerance of ≤30%.

The spatial distribution of the parameters was modeled using the modules RMA2 and RMA4 in SMS. SMS is a modular set of computer programs that simulate surface-water flow that is essentially two-dimensional in a horizontal plane. RMA2 was constructed to provide the hydraulic model, while RMA4 was used to simulate the fate and transport processes of pollutants in the lake and to investigate physical processes responsible for the distribution of pollutants in the lake. The model is limited to solving one- and two-dimensional problems. Modules RMA2 and RMA4 are currently implemented by the US Army Corps of Engineers [24]. The selection of this model was based on the fact that Lake Chapala is a shallow freshwater body [19] and, thus, can be treated as a 2D-system. The equations of the mass and momentum balances were solved using the routines RMA2 and RMA4 in the SMS software. In both models, the hydraulic parameters were adjusted according to the field-collected data. To verify the precision of the models, the calculated concentrations of pollutants were compared with the measured concentrations.

2.6.1. Governing Equations

RMA2 is a module that solves a 2D finite element hydrodynamic numerical model that is depth-averaged [25]. The computations include water surface elevations and components of the horizontal velocity for subcritical free-surface two-dimensional flow fields. The Reynolds form of the Navier-Stokes equations is solved using RMA2 for turbulent flows [26]. Manning's expressions are used to incorporate friction effects, whereas turbulent features are defined using eddy viscosity concepts. The Galerkin finite element numerical solutions obtained in RMA2 incorporate one- and two-dimensional quadrilateral or triangular elements; quadratic functions are used for velocity, and linear functions are used for depth. RMA2 is designed to solve situations with insignificant vertical accelerations and velocity vectors pointing in the same direction throughout the full depth of the water column. The hydraulic model is derived from a momentum balance performed on a control volume of the lake, assuming that the fluid is incompressible and of constant density [26], as shown in Equation (A):

Rate of accumulation in control volume = Rate of momentum into control volume − Rate of momentum out of control volume ± Sum of forces acting on control volume          (A)

The three equations are the $x$ and $y$ momentum equations and the continuity equation, given by the following expressions and the corresponding boundary conditions (BC's):

$$h\frac{\partial u}{\partial t} + hu\frac{\partial u}{\partial x} + hv\frac{\partial u}{\partial y} + \frac{h}{\rho}\left(E_{xx}\frac{\partial^2 u}{\partial x^2} + E_{xy}\frac{\partial^2 u}{\partial xy^2}\right) + gh\left(\frac{\partial a}{\partial x} + \frac{\partial h}{\partial x}\right) + \left(\frac{gun^2}{1.486h^{\frac{1}{6}}}\right)^2 +$$
$$(u^2 + uv^2)^2 - \zeta V_a^2\cos\psi - 2h\omega sinv\varphi = 0 \tag{1}$$

BC's:

$$u(x,y,0) = constant \; for \; t = 0 \tag{2}$$

$$u(x,y,t) = Q_{Lerma} \Big/ Ac_{Maltaraa} \approx 0.06 \tag{3}$$

$$when \; 75,270 \leq x \leq 75,282; \; 1403 \leq y \leq 1452; \; 0 \leq t \leq 3.15 \times 10^7 \tag{4}$$

$$u(x,y,t)_{exit} = f(h), \qquad 0 \leq t \leq 3.15 \times 10^7 \tag{5}$$

when $x = 30,453$; $y = 3979$

$$h\frac{\partial v}{\partial t} + hu\frac{\partial v}{\partial x} + hv\frac{\partial v}{\partial y} - \frac{h}{\rho}\left(E_{yx}\frac{\partial^2 v}{\partial x^2} + E_{yy}\frac{\partial^2 v}{\partial y^2}\right) + gh\left(\frac{\partial a}{\partial x} + \frac{\partial h}{\partial x}\right) + \left(\frac{gvn^2}{1.486h^{\frac{1}{6}}}\right)^2 + \quad (6)$$
$$\left(u^2 + uv^2\right)^2 - \zeta V_a^2\cos\psi - 2h\omega\sin v\varphi = 0$$

BC's:

$$v(x,y,0) = constant, \ t = 0 \ for \ (x,y) \in Dv(x,y,t) \ Maltaraa \approx 0 \tag{7}$$

$$when \ 75,270 \le x \le 75,282; \ 1403 \le y \le 1452; \ 0 \le t \le 3.15 \times 10^7 \tag{8}$$

$$v(x,y,t)_{exit} \approx 0 \tag{9}$$

$$when \ x = 30,453; \ y = 3979; \ 0 \le t \le 3.15 \times 10^7 \tag{10}$$

$$\frac{\partial h}{\partial t} + h\left(\frac{\partial u}{\partial x} + \frac{\partial v}{\partial y}\right) + u\frac{\partial u}{\partial x} + v\frac{\partial h}{\partial y} = 0 \tag{11}$$

BC's:

$$h(x,y,0) = constant, \ t = 0 \tag{12}$$

$$h(x,y,t) = f(t) \approx constant = h_{Maltaraa} = 2.75 \tag{13}$$

$$when \ 75,270 \le x \le 75,282; \ 1403 \le y \le 1452; \ 0 \le t \le 3.15 \times 10^7 \tag{14}$$

$$h(x,y,t) = f(t) \approx constant = 1.5; \ 0 \le t \le 3.15 \times 10^7 \tag{15}$$

$$when \ x = 30,453; \ y = 3979 \tag{16}$$

In the equations above:
$h$ = lake water column depth, m.
$u, v$ = velocities in the cartesian directions, m/s.
$t$ = time, s.
$\rho$ = water density, kg/m$^3$.
$E$ = Eddy viscosity coefficient: $xx$ = normal direction on the $x$ axis surface; $yy$ = normal direction on the $y$ axis surface; and $xy$ and $yx$ = shear direction on each surface, kg/m·s (or Pa·s).
$g$ = acceleration due to gravity, 9.81 m/s$^2$.
$a$ = elevation of lake bottom with respect to the maximum level of the lake, m.
$n$ = Manning's roughness n-value coefficient,
1.486 = conversion factor from SI (metric) units to non-SI units.
$\zeta$ = empirical wind shear coefficient, dimensionless.
$V_a$ = wind speed, m/s.
$\psi$ = wind direction, degrees.
$\varphi$ = local latitude, degree.
$h_{Maltaraña}$ = average depth of the Lerma River at the entrance of the lake, m (see Figure 1).
$Q_{Maltaraña}$ = Lerma River flow rate at the river mouth to the lake, m$^3$/s.
$A_{C \ Maltaraña}$ = cross-sectional area of the Lerma River at the entrance of the lake, m$^2$.

Equations (1), (6) and (11) correspond to the modified Navier-Stokes equations [26], developed by the US Army Corps of Engineers team [11], and consider the wind action and Coriolis force affecting the hydrodynamic parameters of water bodies.

The RMA4 module includes a model that simulates the fate and transport processes of pollutants in a water body, such as a lake. The model is limited to solving one- and two-dimensional problems. RMA4 includes a water quality transport numerical model with finite elements. The main assumption is that the pollutant concentration distribution in the vertical dimension is assumed to be uniform. The equation for this model is derived from the mass balance of any pollutant in Lake Chapala. This balance considers advective and dispersive processes, sorption-desorption phenomena at equilibrium (e.g., metals' partition into dissolved and particulate species), sedimentation and resuspension, diffusion through the water-sediment interface, and the action of wind on the surface of Lake Chapala. A mass balance in any particular region of the lake is:

$$Accumulation = inputs - outputs \pm kinetics \pm transport\ processes \qquad (B)$$

where the term accumulation refers to the change in concentration of a pollutant (metal or nutrient). The inputs of pollutants represent external sources, including possible inputs into the lake by any means, such as wastewater treatment plant discharges, rivers, or runoffs. The outputs refer to any point in the lake from which water is extracted. In this study, the Lerma River was considered the only important inlet source of pollutants moving into the lake, and site S6 was considered the only outlet of pollutants from the lake. The natural drain of the lake was not considered since Lake Chapala does not overflow due to its low volumes. The last term of Equation (B) considers the mass transport processes from the water column to sediment, and vice versa, for any pollutant. The depth-integrated equation of the transport and mixing processes is as follows:

$$h\left( \frac{\partial C}{\partial t} + u\frac{\partial C}{\partial x} + v\frac{\partial C}{\partial y} - \frac{\partial}{\partial x}D_x\frac{\partial C}{\partial x} - \frac{\partial}{\partial y}D_y\frac{\partial C}{\partial y} + \sigma + \frac{R(C)}{h} + \Omega \right) = 0 \qquad (17)$$

BC's:

$$C(x,y,0) = C_o\ for\ (x,y) \in D \qquad (18)$$

$$C(x,y,t) = C_{Lerma}\ when\ 75,270 \le x \le 75,282;\ 1403 \le y \le 1452;\ 0 \le t \le 3.15 \times 10^7 \quad (19)$$

$$C(x,y,t) = C_{exit}\ when\ x = 30,453;\ y = 3979;\ 0 \le t \le 3.15 \times 10^7 \qquad (20)$$

In the equations above:
$h$ = lake water column depth, m.
$C$ = concentration of pollutant for a given constituent, $g/m^3$.
$t$ = time required to apply the mass balance, s.
$u, v$ = velocities in $x$ and $y$ directions, m/s.
$D_x, D_y$ = turbulent mixing (dispersion) coefficient, $m^2/s$.
$\sigma$ = source/sink of constituent concentration, $g/m^3 s$.
$R(C)$ = rainfall/evaporation rate, $g/m^3 s$.
The term descriptions are as follows:
1st term = local storage.
2nd term = advection term ($x$).
3rd term = advection term ($y$).
4th term = dispersion ($x$).
5th term = dispersion ($y$).
6th term = local sources of mass substances.
7th term = rainfall/evaporation effects.

8th term = term that includes sorption-desorption, sedimentation, resuspension, and partition processes, g/m$^3$s. $\Omega$ is taken as a term to model the transport of particulate or dissolved metals by adsorption/ desorption, and point sources/active layer of sediment. $\Omega$ is given by the following equations:

(a) For the simulation of particulate metals [27]:

$$\Omega = -\sum K_i\left(C_{pi} - K_d C_{di} C_{si}\right) - \frac{v_s}{h} C_{pi} + \frac{v_r}{h}\left(C_{pi}\right)_B + S_{pi} \tag{21}$$

(b) For the simulation of dissolved metals [27]:

$$\Omega = \sum K_i(K_d C_d C_{si}) - \sum K_i \gamma_i (1 - \varnothing)\frac{D_i}{h} K_d C_d + \sum K_i \gamma_i (1 - \varnothing \gamma)\frac{D_i}{h} r_2 + S_d \tag{22}$$

In the equations above:

$S_p$ = the level of concentration, i.e., how diluted, or how concentrated the source of the pollutant is, g/m$^3$·s.

$K_i$ = rate of adsorption/desoprtion that will tend to reach equilibrium with sediment, s$^{-1}$.

$K_d$ = partition coefficient between particulate metal and dissolved metal, m$^3$/g.

$C_{si}$ = suspended mass associated with the sediment, g/m$^3$.

$\varnothing$ = porosity of the sediment.

$D$ = average particle diameter in the sediment, m.

$r_2$ = concentration of particulate metal associated with the active layer of sediment, g$_{metal}$/g$_{sediment}$.

$S_d$ = mass balance between metal consumption from the aqueous phase toward the sediment and detachment from the sediment phase toward the aqueous phase caused by an advective effect on the element, g/m$^3$·s.

$\gamma$ = g/m$^3$.

### 2.6.2. Model Setup

The solution of Equations (17)–(22) often requires the previous solution of the hydraulic model Equations (1)–(16), in the pollutant modeling of water bodies, which is called coupled modeling [28,29]. Since Lake Chapala is a shallow lake, it follows the well-mixed regime, and the previous equations can be applied. The solution of these models requires the creation of a mesh, through which a system of non-linear simultaneous algebraic equations is generated and then solved using the Newton-Raphson method. To create the mesh, the bathymetry of the lake was required (Figure 2a). Equations (1)–(21) or (1)–(22) were solved simultaneously. RMA2 was first solved for each finite element in the hydraulic model, and the solutions of $h$, $u$, and $v$ were used by RMA4 to solve for the transport of pollutants.

### 2.7. Model Calibration Procedures

To arrive at a flow model for Lake Chapala, the bathymetry data of the lake were imported as a set of $x$, $y$, and $z$ data points provided by the Mexican Water National Commission [30]. Figure 2a shows the bathymetry used. With the bathymetry data incorporated, SMS generated a finite element mesh by employing these points as corner nodes. Several meshes were tested in the simulations, and the final mesh was obtained through a refining process and modification of the geometric elements of the mesh. The resulting mesh contained 6357 geometric rectangular and triangular elements and a total of 17,068 nodes (Figure 2b).

**Figure 2.** (**a**) Lake Chapala bathymetry. Mi = Mezcala Island; Ai = Alacranes Island. The bathymetric heights are above mean sea level (AMSL) [30], and (**b**) a finite element mesh was used in the present study.

The calibration of the hydraulic model parameters included the Eddy viscosity coefficients $E_{xx}$ and Manning's roughness coefficient $n$. Due to the lack of experimental data for $E_{xx}$ and $n$ in Equations (1)–(16), the calibration of the hydraulic model was performed by a trial-and-error iterative process. According to Hosseiny [31], trial-and-error methods are one of three major categories of approaches for optimization in model calibrations. First, the values of the viscosity coefficients $E_{xx}$ given as defaults in the software were used; then, these coefficients were varied until the values of $h$ and $v$ approached the measured values within an acceptable error tolerance. The lake was divided into 12 zones so that each zone contained at least one site, from which the water sample was taken and a velocity measurement was performed. To each zone, we assigned values of the dispersion coefficients $D_x$ and $D_y$. In the case of the eddy viscosity coefficient, $E_{xx}$, an isotopic value was assigned, i.e., a mixed value containing the same magnitude in both directions (*x* and *y*, respectively), with the aim of simplifying the model and avoiding divergence. Both types of coefficients were adjusted by trial and error until the simulated variables of the two models approached the measured variables.

*2.8. Model Validation*

The model validation was performed using the relative percentage difference (RPD) parameter [32,33], given by: RPD $= \dfrac{X_{sim} - X_{meas}}{X_{meas}} \times 100$, where $X_{sim}$ is the simulated variable and $X_{meas}$ is the measured variable. Positive results designate model underestimations, and negative results represent model overestimations. An acceptable tolerance value criterion was established through the RPD, set to $\leq 30\%$, since Lake Chapala has a large surface area and low depth ($\leq 7$ m) that render it sensitive to constant changes in the hydraulic variables due to wind action over the surface [34]. Additionally, the velocity directions were used to validate the model by changing the values until they were close to those reported by Filonov [34]. Once the hydraulic model was calibrated, both the transport and hydraulic models were solved simultaneously. Due to the simplified model boundary conditions and neglected biogeochemical cycling of nutrients, the validation of the model using the RPD can result in statistical biases. As a result of this, this model is useful as a first approximation, particularly in bi-dimensional systems such as the shallow lake Chapala. To decrease the level of uncertainty, it is advisable to combine it with a statistical method and verify the distribution of the data. For instance, one can combine the RPD with the method recommended by the World Metrological Organization, whose accuracy

(uncertainty levels) expressed at the 95% confidence interval is ±5% [35]. However, this is outside the objectives of this work and will be presented in a future work.

## 3. Results and Discussion

### 3.1. Hydraulic Model Calibration Results

Figure 3 depicts a scaled velocity vector map. The lack of simulated vectors at the Lerma River mouth (east side of the lake) is due to the low water volumes of the lake. It can be observed that high velocities at the Lerma River entrance and in the zones around sites S8 and S9 and the outlet of the lake create large vectors that obscure and block the determination of the details of these zones. However, the velocity contour solutions result in a mixing pattern of the waters of the lake. The measured field data of the water depth (Figure 4a,b) and velocity (Figure 4c,d) were compared with the corresponding simulated data to verify the prediction of the hydraulic model. The range of simulated depths was 0.7–7.1 m, whereas the water movement velocities were between 0 and 0.08 m s$^{-1}$. The RPD of the depth did not exceed 25% at any of the sites in either of the two sampling campaigns, with better results shown for the 2014 campaign (Figure 4a). In the case of the water movement velocities, the model predicts RPD values < 30% for all sites in the two sampling campaigns (Figure 4c,d), except for sites S9 and SE6. Site S9 has a negative calculated velocity, indicating that the water movement occurs in the opposite direction to the measured registered value. The differences in the calculated velocities and directions at these two sites may be due to the fact that the hydraulic processes taking place in these zones are more complex than those considered by the model.

### 3.2. Calibration Results of Model Coefficients

Tables S2 and S3 (Supplementary Information) depict the values of the dispersion coefficients used for the calibration of the transport model and the values of the transport of nutrients and metals. In these tables, the DVFs (dominant velocity fields) tell the program that the dispersion of pollutants changes direction according to the velocity fields for the given zone. In the case of nutrients, it is observed that the dispersion coefficients range between $1 \times 10^{-4}$ and $10 \times 10^{-4}$ m$^2$s$^{-1}$, whereas in the case of metals, the coefficients range between $0.5 \times 10^{-4}$ and $9 \times 10^{-4}$ m$^2$s$^{-1}$. The values of these coefficients are close to those reported in the literature [36,37]. It can also be observed that the dispersion coefficients of nutrients are larger than those of metals. This indicates that nutrients are more easily dispersed throughout the lake. This has a significant effect on the presence of these pollutants in that the water quality of the lake is more affected by nutrients than by metals. The sources of these nutrients include inputs from the Lerma River and agricultural runoffs from cultivated lands along the lake [38]. The results of the eddy viscosity coefficient $E_{xx}$, applied to the calibration of the hydraulic model, fall within the range of 47.5–49.5 Pa·s and remain almost invariant for a given zone in the two sampling campaigns. The values of this coefficient fall within the characteristic range of shallow water bodies with low motion [24]. It can be concluded that advective and dispersive processes are involved in the transport of metals, whereas dispersive processes predominate in the transport of nutrients, with a significant impact on the water quality of the lake.

### 3.3. Pollutant Simulations

#### 3.3.1. Nitrates

The nitrate simulations (2014) are shown in Figure 5A as contour curves. Figure 5A(d) shows that the entrance zone of the Lerma River is where the concentrations are the highest in a 12-month simulation (1.05–1.13 mg L$^{-1}$). The Lerma River collects municipal wastewaters rich in nitrogen and nitrates, with Lake Chapala as their destination [14]. In general, it can be observed from Figure 5A(a–d) that nitrates tend to disperse from east to west and either maintain an almost constant concentration or diminish. This is attributed to kinetic processes of the nitrogen cycle that consume important amounts of nitrates. Nitrates are part of a complex cycle that includes reaction kinetics, i.e., a cycle whereby nitrate ions

are generated and consumed simultaneously. A deficiency of either organic matter or microorganisms can produce a nitrate concentration that remains constant with respect to one of the inlets. Figure 5B shows the results of the simulations for the second sampling campaign. The nitrate concentration does not change in the 1-month simulation (Figure 5B(a)) compared with the initially measured value (~0.9 mg L$^{-1}$), whereas significant changes can be observed in Figure 5B(b,c). It can also be observed in Figure 5B(b–d) that there is an increase in the simulated nitrate concentration between the Lerma River entrance and the central part of the lake, indicating that there is a significant generation of nitrates in these zones. This could be due to both the production of ammonium ions (NH$_4^+$), which originate from excretions of aquatic fauna, and the decomposition of organic matter from algae or aquatic microbiota, which release ammonia, which later undergoes conversion into ammonium ions and ultimately into nitrates under the action of bacteria [39,40].

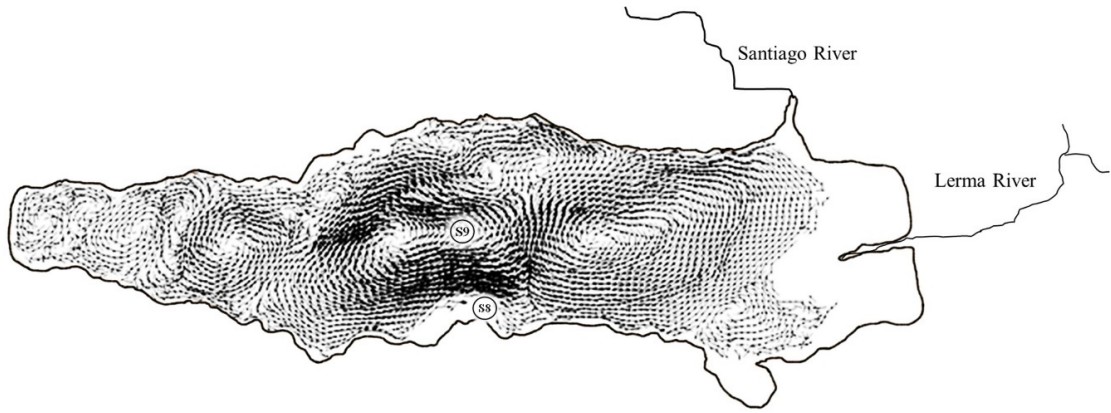

**Figure 3.** Movement patterns and velocity vectors of Lake Chapala.

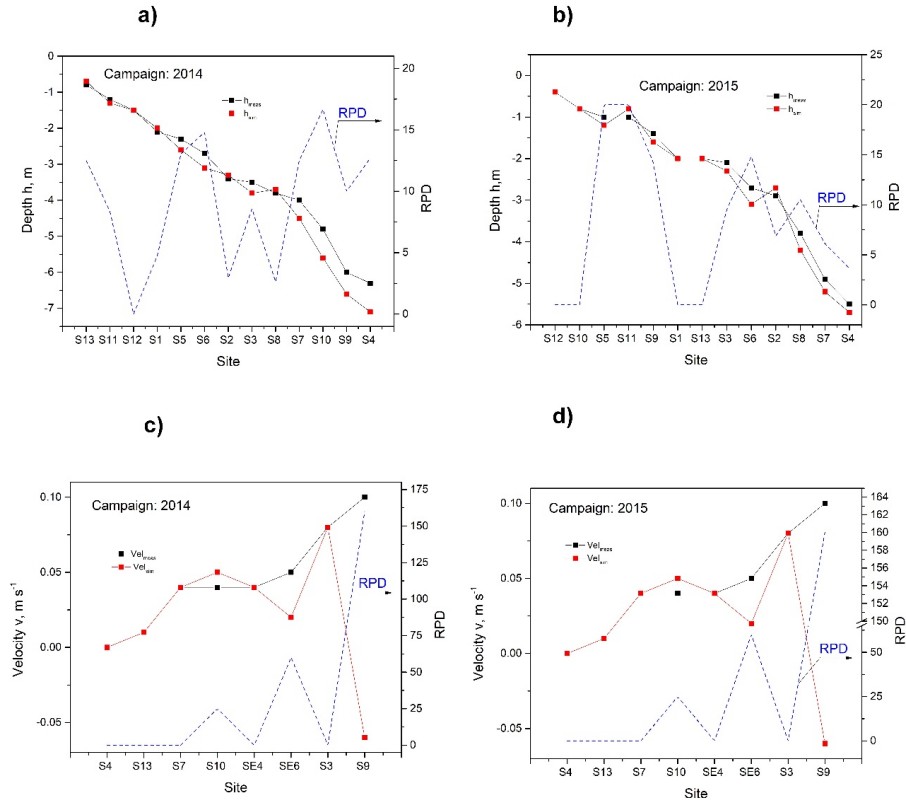

**Figure 4.** Validation of the hydraulic model of Lake Chapala: (**a**) depth, 2014; (**b**) depth, 2015; (**c**) velocity, 2014; (**d**) velocity, 2015. meas = measured; sim = simulated.

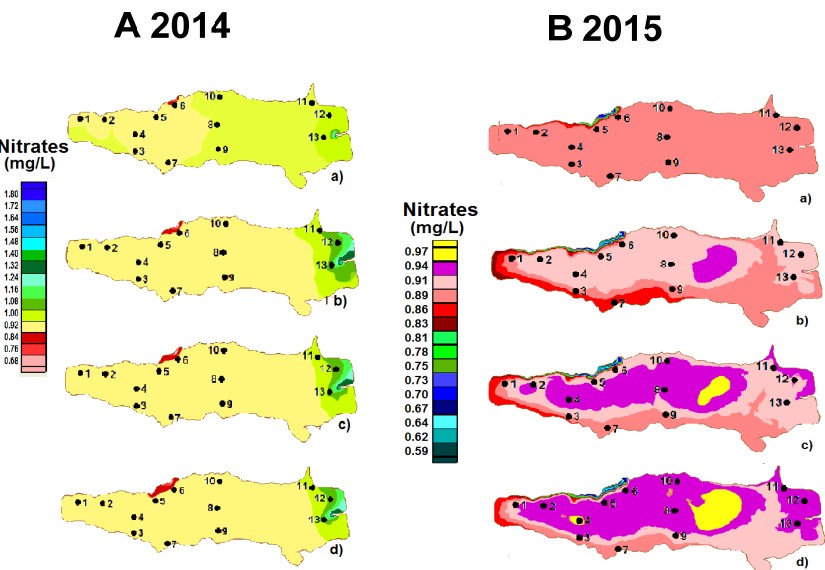

**Figure 5.** Depth averaged contour graphs for nitrates. (**A**): 2014; (**B**): 2015. (**a**) Month 1, (**b**) Month 4, (**c**) Month 8, (**d**) Month 12.

The results of the simulations and RPD values of nitrates for the two sampling campaigns are shown in Figure 6. The RPDs obtained in the simulations of this nutrient correspond to the 12-month simulation period. The reason for choosing this period for this calculation was to observe the behavior of the model over long periods, which could aid the managing authorities of Lake Chapala in making decisions regarding water quality issues in terms of the nutrients that can cause algae to flourish. It can be observed from Figure 6a that the RPD results are low for all the sites except for site S7 and that the calculated concentrations, in the case of the 12-month simulations, range between 0.93 and 0.13 mg L$^{-1}$. The RPD values shown in Figure 6a are satisfactory for most of the sites except for site S7, which depicts the most significant variation in the concentrations for the 12-month simulations. This finding can be attributed to wastewater discharge types that are not incorporated into the model and omitted point and non-point sources, such as runoffs from nearby cultivated lands as well as mountains close to the lake, which have been subject to erosion processes due to deforestation and land use, constituting the principal non-point nutrient source for the lake [38]. The presence of these agrochemicals finally increases the concentrations of nutrients in the lake. However, these point and non-point sources are difficult to assess accurately; therefore, only approximations can be obtained. When these sources were included in the simulation model, they generated inconsistencies. Additionally, sites S11 and S12 present the highest calculated concentrations for the simulations of 4, 8, and 12 months; the RPD values for these sites do not exceed 30%. A close inspection of the data in Figure 6b reveals that at sites S11 through S13, the calculated concentration is increasing, whereas the opposite trend is observed at sites S1 and S5, where the calculated concentration initially decreases but then, at the end of the simulation, increases above the measured concentration. In the 12-month simulation, the nitrate concentration ranges between 0.776 and 0.968 mg L$^{-1}$, and site S4 has the highest concentration.

### 3.3.2. Phosphates

Figure 7A,B depict the results of the phosphate simulations as a contour map for the two sampling campaigns. In the case of the first sampling campaign, Figure 7A(b) presents two zones (center and east), which are separated by a considerable distance, and both zones present the same calculated concentration (1.1 mg L$^{-1}$). This can be explained based on a hydraulic simulation, in which it was considered that a particle moves according to the velocity and direction of the water movement in the lake (Figure S1). Figure S1a shows the transport of one particle from the moment when it enters the lake up to the

moment when it leaves. The white colored lines represent the particle's path, and the dark colored background represents the sites through which the particle is very unlikely to pass. Figure S1b shows the same process, except that here 10 particles are considered. This simulation indicates the formation of a vortex (swirl) in the water for some zones, as indicated by arrows. There are zones shown in black that exist in the centers of these vortexing structures and through which the particle is very unlikely to move. This situation can be compared to the case where a centrifuge force pushes a particle outward in a circular motion. This physical situation will cause this particle's concentration to be higher at the banks than at the center of the vortex. By applying this rationale to the lake, if this process takes place as explained above, it will affect the phosphate concentrations since these substances tend to sorb easily onto fine, solid suspended matter [41]. A close inspection of Figure 7A(b) reveals that one of the two previously mentioned zones is within the vortex zone (Figure S1). This helps to explain why zones with low phosphate concentrations can be observed as patches in the lake (Figure 7A,B). Figure 7B(a–d) present zones with almost homogeneous simulated phosphate concentrations. From the simulations shown in Figure 7A(d), it can be observed that the calculated concentrations range between 1.55 and 2.36 mg L$^{-1}$, corresponding to sites S12 and S6, respectively.

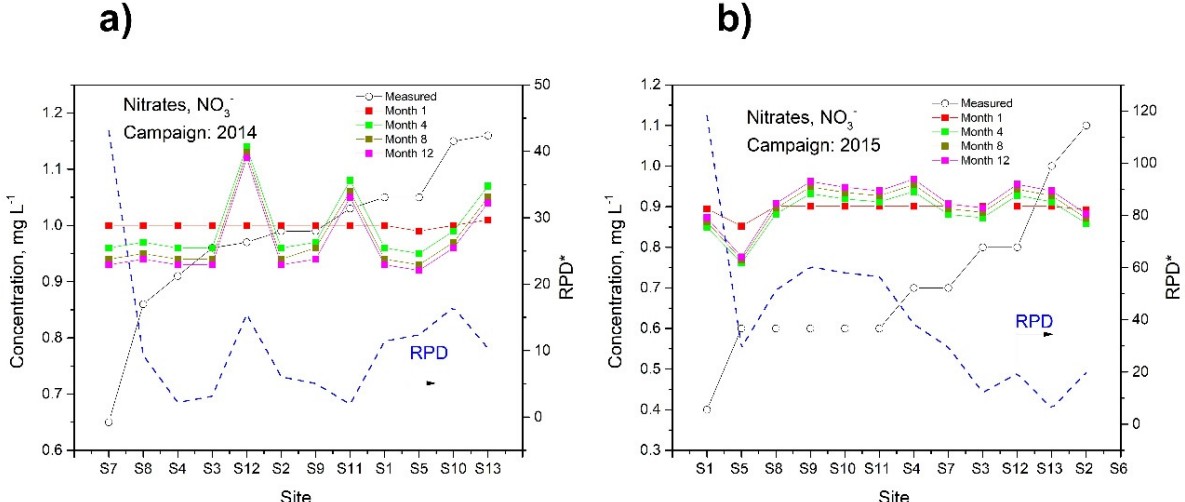

**Figure 6.** Simulation and RPD results for nitrates: (**a**) 2014, (**b**) 2015. * RPD calculated with respect to month 12.

Figure 8a,b show acceptable RPD values for the phosphate simulations of all the sites. Differences can indicate that other factors (such as biological processes or higher-order kinetic terms) should be incorporated into the model to obtain improved simulations. In the case of the second sampling campaign, high phosphate concentrations were calculated (Figure 8b). An explanation for these high phosphate concentrations can be derived from Figure S1, where the model predictions indicate that the phosphates move from east to west in a trend of displacement towards the northern part of the lake. This phenomenon could be the reason why sites S10, S12, and S13 present the highest concentrations. The RPD obtained for this campaign (Figure 8b) indicates that the simulations are not satisfactory since they are not close to the measured values. It has been reported that the greater the $PO_4^{3-}$ leaving the lake is in comparison with the incoming sources, the more the lake will behave as a source of $PO_4^{3-}$ due to the internal load derived from the sediments [42]. If the amount of $PO_4^{3-}$ leaving the lake is small compared with the incoming amount, this indicates that the lake acts as a sink for $PO_4^{3-}$, e.g., by burial in the sediments [43]. De Anda and Maniak [42] also reported that the lake can either remove or accumulate these incoming nutrients. They found that the lake showed a capacity for removing $PO_4^{3-}$ just before the year 1983. After this period, the lake was not able to remove this nutrient by a depuration process, and it has continued to accumulate $PO_4^{3-}$ in both the water column

and the sediments ever since. These events can help explain the lack of predictions made by the model since these factors were not considered in the simulation. A study is in progress, aiming to incorporate these events into the simulation.

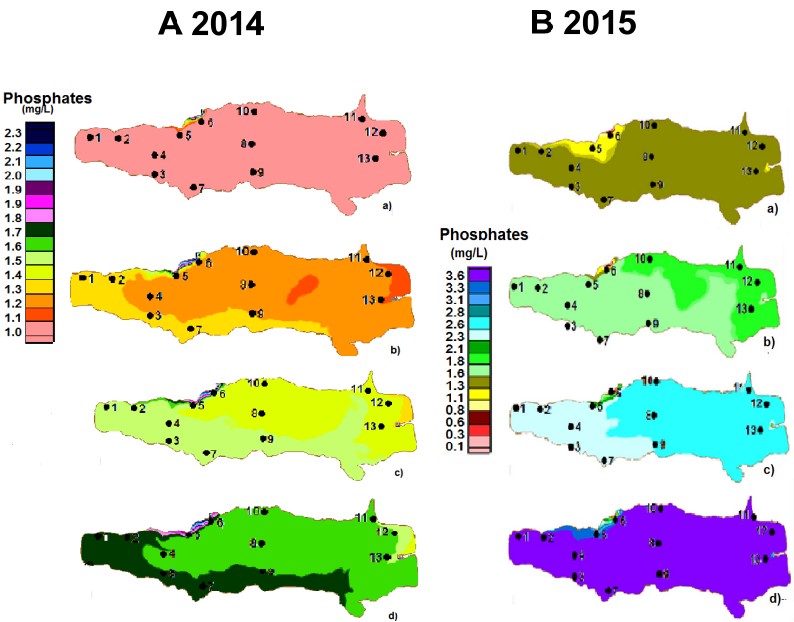

**Figure 7.** Depth-averaged contour graphs for phosphates. (**A**): 2014; (**B**): 2015. (**a**) Month 1, (**b**) Month 4, (**c**) Month 8, (**d**) Month 12.

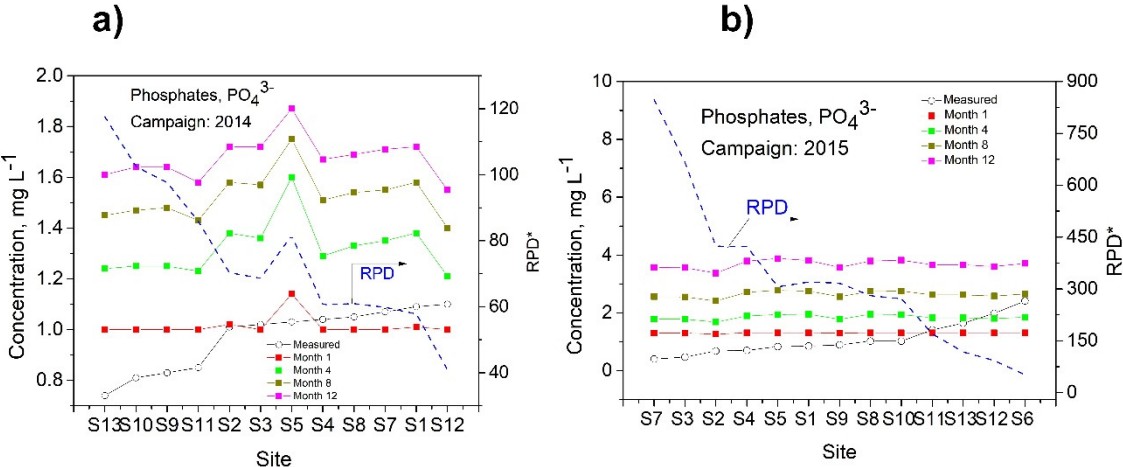

**Figure 8.** Simulation and RPD results for phosphates: (**a**) 2014; (**b**) 2015. * RPD calculated with respect to month 12.

### 3.3.3. Nickel

The concentrations of Mn, Cr, Cd, Cu, Pb, and Fe for the two sampling campaigns were all below the limit of detection of the instrument. Here, we present the results for Ni and Zn only. In the sampling campaign of 2015, the concentrations of $Ni_{total}$ and $Ni_{diss}$ were all below the limit of detection of the instrument; hence, no simulations were performed in this campaign. Figure 9A shows contour graphs for the Ni particulate ($Ni_{part}$). They indicate initial concentrations of $Ni_{part}$ of 0.05–0.06 $mgL^{-1}$, whereas in the next 8-month simulations (b and c in Figure 9), the concentrations vary between 0.09 and 0.171 mg $L^{-1}$. In the 12-month simulations, the eastern zone presents the highest concentration range (0.238–0.255 $mgL^{-1}$). At site S6, the concentration variations fall within the same range as those of the 12-month simulations. The presence of these varying concentration profiles

at the inlet of the lake suggests that the Lerma River and resuspension processes that occur in the lake act as potential sources of nickel. The results of $Ni_{diss}$ are depicted in Figure 9B. The simulated values are closer to the measured values compared with those of the first sampling campaign and are close to the detection limit of the instrument. The results shown in the contour graphs indicate a pattern that is practically identical to that of $Ni_{part}$. In this case, the simulated results of the 12-month period suggest that $Ni_{diss}$ moves from east to west and that the eastern zone reflects the greatest concentration variations at sites S11–S13 (Figure 9B(d)), with the rest of the lake showing almost homogeneous concentrations (0.022 mg L$^{-1}$).

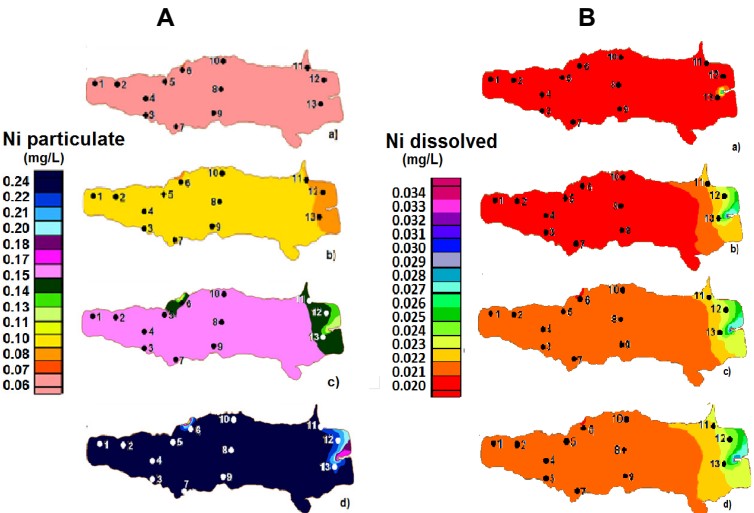

**Figure 9.** Depth-averaged contour graphs for (**A**): $Ni_{part}$, (**B**): $Ni_{diss}$ (2014). (**a**) Month 1, (**b**) Month 4, (**c**) Month 8, (**d**) Month 12.

The RPDs are depicted in Figure 10a,b. The RPDs observed in Figure 10b indicate a good approximation of the model, with values of ≤39%, despite the low measured concentrations of nickel. In summary, the model predicts $Ni_{diss}$ to a satisfactory degree compared with $Ni_{part}$, which can be attributed to the fact that in the former case, resuspension processes that otherwise could interfere with the simulations are absent from the model.

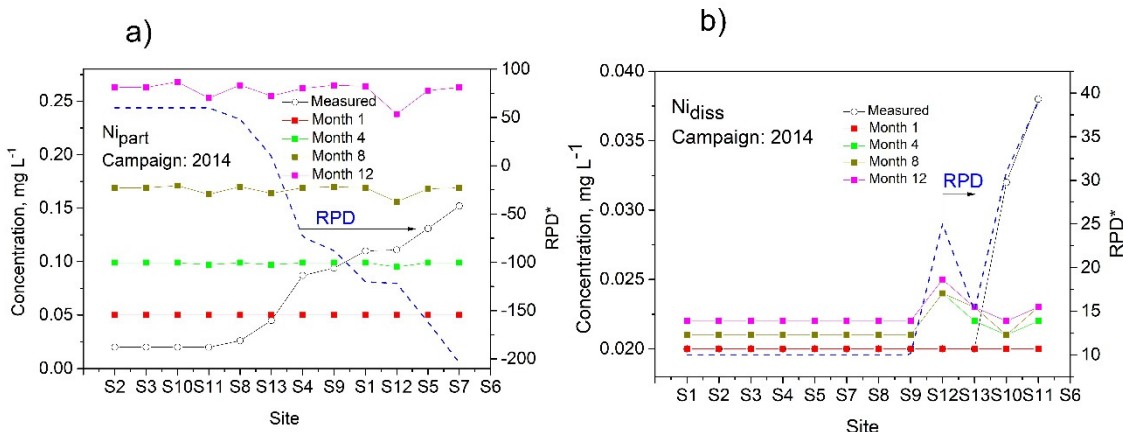

**Figure 10.** Simulation and RPD results for nickel (2014): (**a**) $Ni_{part}$; (**b**) $Ni_{diss}$. * RPD calculated with respect to month 12.

### 3.3.4. Zinc

Zinc. The $Zn_{total}$ results refer to the concentrations detected by the instrument during the 2015 campaign, while in the case of the sampling campaign of 2014, the concentrations

of $Zn_{total}$ and $Zn_{diss}$ were all below the limit of detection of the instrument; hence, no simulations were performed for this campaign. The concentration simulation results are shown in Figure 11A. $Zn_{total}$ shows increases of 0.04 mgL$^{-1}$ from the initial simulated concentrations on average for the 4-month simulated period (Figure 11A(a)). In the case of the 8- and 12-month simulated periods (Figure 11A(c,d)), the concentrations increase by 0.06 and 0.11 mgL$^{-1}$, respectively. Regarding the concentration differences from one site to another in a given simulation period (e.g., the 12-month period), the highest concentration differences are observed in the eastern zone, but in contrast to the Ni simulations, the rest of the lake's zones present small simulated concentrations (Figure 11A(d)). These results indicate that the Lerma River is a source of zinc. The RPD is shown in Figure 11B, where the highest values are reported for site S5, followed by sites S9 through S12, and the modeling simulations exceed 30%. These differences may be due to particle resuspension processes not considered by the model that take place in the waters of the lake at these sites due to the presence of the vortex detected in the hydraulic model simulations [42].

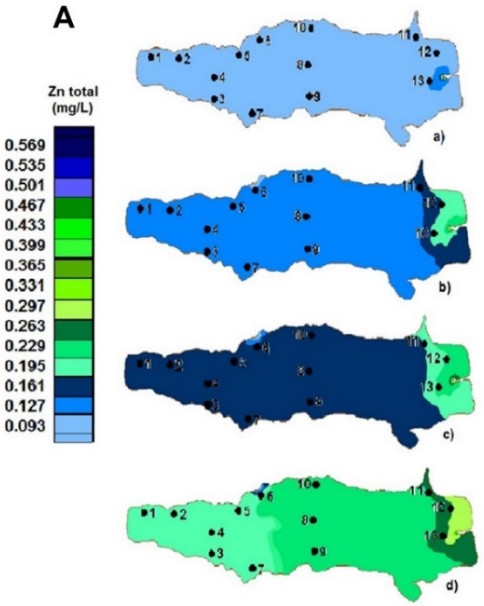

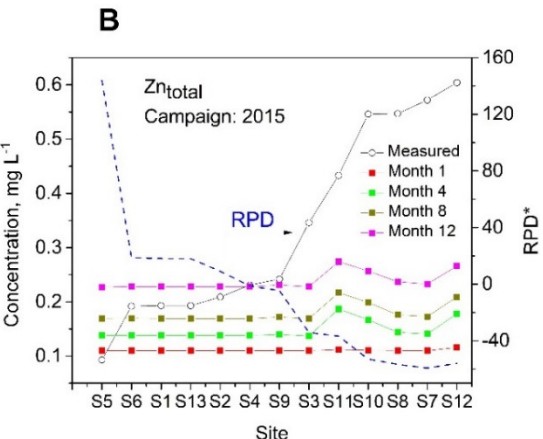

**Figure 11.** (**A**): Depth averaged contour graphs for (**A**): $Zn_{total}$, (**B**): simulation and RPD results (2015). (**a**) Month 1, (**b**) Month 4, (**c**) Month 8, (**d**) Month 12. * RPD calculated with respect to month 12.

## 4. Discussion

The results of the mathematical model were discussed to explain a 3D system using a bidimensional one, but the model may be subject to uncertainties. As a result of this, the

model is useful for obtaining a first approximation, but it must be considered that there are alternatives for assessing biases, which can be handled using multivariate statistical methods, such as factor analysis, which could help to explain the variability in the measured and simulated data in addition to providing trust limits.

The results of this model, which we applied to Lake Chapala, demonstrate the problem in which high levels of nutrients cause the eutrophication of the main source of water supply to the city of Guadalajara, Mexico. These results can be useful for predicting scenarios involving unsafe water, helping decision-making authorities implement stricter measures that could contribute to the maintenance of this ecosystem's health, including the following: (a) surveillance of the types of industrial water discharge into the lake; (b) investment in the installation of more wastewater treatment plants in towns and cities located along the shore of the lake and at the inlet of the Lerma River; and (c) the enforced application of fertilizers and agrochemicals that are environmentally friendly and capable of rapid degradation by farmers whose lands are located along the shore. The implementation of these measures could contribute further to the generation of information, helping to achieve compliance with the Sustainable Development Goals (SDGs) in order to guarantee safe water for all users of this water body, according to indicator 6.3.2.

## 5. Conclusions

A coupled 2D model, including hydraulic and transport simulations, was applied using the RMA4 and RMA2 modules to predict different pollution scenarios for Lake Chapala. The SMS provided a suitable method for the lake model's discretization through the creation of a final mesh and the acquisition of contour maps that proved useful for gaining more thorough insights into the pollution situation of this lake. Overall, the numerical model showed good validation results when analyzed against the water level, current velocity, and pollutant measurement data collected through the RPD, except at some sites and for the long simulation periods. Site S2 showed the highest $NO_3^-$ values, which were attributed to runoff from cultivated lands dragging fertilizers with high nutrient contents into the water. We observed an increase in the simulated nitrate concentration between the entrance zone of the Lerma River and the central part of the lake, indicating that there is a significant generation of nitrates in these zones. The simulation results obtained using RMA2 permitted the identification of vortex locations in different zones of the lake, which influenced the location and concentration levels of phosphates. All the pollutant simulations allowed us to observe, for the first time, zones with greater concentrations at the mouth of the Lerma River, demonstrating that this river acts as the main contributor of nutrients and metals moving into the lake. The calibration results of the hydraulic model indicated that dispersive processes were more significant for nutrients than for metals. This indicated that nutrients are more easily dispersed throughout the lake. This has a significant effect on the presence of these pollutants in that the water quality of the lake is more affected by nutrients than by metals. The sources of these nutrients included inputs from the Lerma River and agricultural runoff from cultivated land along the lake. Runoff from fertilized cultivated lands and sanitary discharge entering Lake Chapala through the Lerma River were the main causes of the elevated concentrations of nitrates and phosphates. It was observed that if these pollutants increased together with a substantial decrease in the amount of dissolved oxygen, they could lead to eutrophication problems in this lake. We highlighted the usefulness of modeling nitrates and phosphates, given that they constitute level 1 pollutants according to indicator 6.3.2 of the SDGs. Modeling can help to maintain the global comparability of this indicator using easy-to-measure water characteristics and represent significant pressures on all parts of the world to ensure the supply of high-quality water for human use.

**Supplementary Materials:** The following supporting information can be downloaded at: https://www.mdpi.com/article/10.3390/w15091639/s1, Figure S1: (a) transport of one particle from the moment it enters until it leaves the lake (b) transport of ten particles from the moment it enters until it leaves the lake. Table S1: Description and GPS location of water sampling sites of Lake Chapala (May 2014 and May 2015). Table S2: Dispersion coefficient values ($D_x$ and $D_y$) of selected contaminants in twelve zones of Lake Chapala (May 2014). Table S3: Dispersion ($D_x$ and $D_y$) and Eddy viscosity Exx coefficient values of selected contaminants in twelve zones of Lake Chapala (May 2015).

**Author Contributions:** Conceptualization, J.O.M.-D., J.B.-C., I.D.B.-Q., E.R.-D. and S.G.-S.; formal analysis, J.I.A.-B.; funding acquisition, S.G.-S.; investigation, J.O.M.-D.; methodology, J.I.A.-B., J.O.M.-D., J.B.-C., P.F.Z.-d.V. and S.G.-S.; software, J.I.A.-B. and S.G.-S.; supervision, S.G.-S.; validation, J.I.A.-B.; writing—original draft, S.G.-S. All authors have read and agreed to the published version of the manuscript.

**Funding:** This research was funded by Mexico's National Council of Science and Technology (CONA-CyT) grant No. 84252 and the Ministry of Public Education-PRODEP grant No. 103.5/13/9346.

**Data Availability Statement:** Not applicable.

**Acknowledgments:** We are thankful to Mexico's National Council of Science and Technology (CONA-CyT) for the scholarships awarded to Jorge I. Alvarez-Bobadilla and Jorge O. Murillo-Delgado.

**Conflicts of Interest:** The authors declare no conflict of interest.

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
