# Peer review of "Modeling Fate and Transport of Nutrients and Heavy Metals in the Waters of a Tropical Mexican Lake to Predict Pollution Scenarios"

_water, doi:10.3390/w15091639_

Round 1
Reviewer 1 Report
In this manuscript, the authors made a trial to model the fate and transport of nutrients (nitrate and phosphate) and metals using the RMA2 and RMA4 modules in the SMS software. I am not familiar with the models used in this manuscript, yet my major concerns are focusing on the model selection, model validation, and the predictive uncertainty (see major comments below). In addition, another major concern about this manuscript is the quality of writing. This paper can be improved in many aspects, including the methods, results and discussion sections (see minor comments below). The English need several rounds of professional language editing.
General comments
A major concern is about the model selection and the predictive uncertainty. Please make it more clear why the models used are approximate for nutrients and metals in this lake, e.g., why a bi-dimensional model is suitable for Lake Chapala, and why other point and non-point sources of nutrients can be neglected in current study. In addition, please add more discussion on the predictive uncertainty of this study, e.g., due to the simplified model boundary conditions, neglected biogeochemical cycling of nutrients, etc.
Another major concern is that some discussion is lack of evidence/references, especially when explaining possible transports or sources of nutrients and metals. For example, line 370-371, line 401-403 (why were NH4+ mentioned here?), 530-532, etc. Please give appropriate references when inferring to probable reasons/sources. Also, thinking how to make a better discussion by analyzing and organizing hydraulic results and nutrient distributions across the lake.
Minor comments
Abstract
Line 18: “were solved” is not approximate here.
Line 20: Unclear descriptions with “good validation results”.
Line 24-25: “Site S2… nutrients”. This sentence is not necessary to be given in the abstract. Has the model considered runoffs of cultivated lands?
Introduction
Line 44-74: This paragraph can be more concise. The authors should summarize the present knowledge and find the gap in former studies instead of just describing the major findings of former studies. In addition, model selection should be introduced, e.g., in this paragraph.
Line 85-87: Unclear description. Please check the grammar of this sentence.
Line 75-84: Please consider moving detailed descriptions of Lake Chapala to Section 2.1, e.g., combine them with line 111-115.
Line 97-104: These descriptions belong to the section 2.
Materials and methods
Line 112-114: Please check the grammar of this sentence. “Is” should be “It is”.
Line 128: Why were both the two sampling campaigns conducted in dry seasons?
Line 133: Table 1 can be moved into the supplementary information.
Line 141: Please mention the analysis methods instead of just using “following protocols established”.
Line 169-174: Please also give the product model and manufacturer name of the instruments mentioned.
Line 176-177: Why were the RMA2 and RMA4 modules in the SMS software suitable for this lake? Descriptions on model selection should be given. Please also mention the group/company that developed this software, or, give approximate references.
Line 178-179: Unclear description with “and it tests for the successful application of remedial control measures at high speed and low cost.”
Line 181-276: Section 2.6.1, please make it clearer where these equations come from and give approximate references. It should also be noted if these equations were directly from the instructions of SMS software.
Line 311: Why did the authors only use RPD during model validation?
Results and discussion
Line 392-395: This is not convincing because denitrification can remove nitrate in the water column through coupled nitrification-denitrification across the sediment-water interface. In addition, denitrification occurring on suspended particles also removes nitrate in the water column.
Line 484-533: Why only results of Ni and Zn were provided in the results and discussion section? The authors said that they measured multiple metals (see section 2.4).
Please add more discussion on the predictive uncertainty of this study, e.g., due to the simplified model boundary conditions.
Conclusion
The conclusion section is quite long and can be more concise!
Reviewer 2 Report
Modeling Fate and Transport of Selected Chemical Pollutants in Waters of a Tropical Mexican Lake to Predict Pollution Scenarios, is an interesting manuscript written referring an important lake.
In the title, instead chemical pollutants, why not use “nutrients and heavy metals”?
Abstract needs improvements: (1) be quantitative (e.g. L22, some sites? At least say ??% of site; long simulations?) (2) L24: nutrient dispersion more important means? (the statement is too broad); (3) L25, S2 (how come the reader knows what is S2)- abstraction is the part where someone reads before deciding to read the manuscript or not (4) Nidiss (do not use abbreviations in abstract unless it is explained) 5) Briefly state the importance of Lake Chapala
Introduction briefly explain, impacts of heavy metals and nutrients. Else, these are just like some modelling variables.
Also, brieftly explain, give details of key aspects, e.g. SMS software (at least you could have used the full term, instead using the abbreviation – surface water modelling system
Figures, clarity not so good
L131: why specifically sampled at vortexes?
Usually at any places tables and figures are written as Tables and Figure (first letter capital)- check the journal style
Table 1: R9 etc (explain what these means), confused as these are stated with some location names? The paper needs to be self-explanatory and rich with all details
L141: NO3^-1, etc use the equation tool to type these
L154: these metal names and nutrients considered, should be included/mentioned near objectives (perhaps in brackets)
L155: Did authors measure total heavy metals? (e.g. Cr III or Cr VI or both)
L161: what you meant by unacidified lake water? Just lake water or, you have made it to be > 7 (pH)?
Did you use a graphite furnace in AAS?
L163-164: Give a reference for the method you adopted for particulate heavy metal concentrations? Usually this gives a high error, isn’t it?
L171: give model details of all equipments (e.g. here, the flow/velocity meter)
Section 2.6.1 give references as and when needed
L293: what was the mesh size?
Fig 2: state the heights (stages) are in AMSL? (at least mention meters?)
L352-357: should be under methodology. Check this throughout the manuscript (also how these 12 zones decided? A simple grid based?)
3.3 “results” part not needed
L385: give references (e.g. here the fact that Chapala is used as an expeidient dump area of wastewater)
Fig 5: Need to improve the hot spots (or special places) that distinctively show up in these diagrams e.g. a-b, the red color area on northern side (mid of the lake at northern boundary)
Separate results and discussion. Currently it includes (over 80%) results. It is fine discussion to be small, but need to have a dedicated discussion, discussing results, reasons and implications. Also it is better to explain how the results indicate the impact of it current status quo in regional/global climate
Suggest, based on results what you could do? E.g. a treatment facility near the river inlet? Some ecological engineering methods?
Good luck
Round 2
Reviewer 1 Report
In the revised manuscript, I think the authors have addressed most of my concerns. Yet, in the response letter, the authors mentioned that "The reason why the results of Ni and Zn were presented only, was because the concentrations of the other metals analyzed were below the limit of detection of our instrument and no simulations could be performed." Please double check the method used (e.g., detection limits) since many of the mentioned metals are undetectable. If possible, the authors can give some explanation on why this method have many undetectable results.
Author Response
Response:
We appreciate the reviewer´s observation. We have double checked the limits of detection of the instrument for Flame Atomic Absorption and they are correct. We also checked these limits with those provided by the APHA manual (Standard Methods for the Examination of Waters and Wastewaters, 2012) and found that they are very much the same.
We believe that the reason why this method has many undetectable results is because the concentrations of the undetected metals were indeed so low to be detected by this technique. The results of the undetected metal concentrations are consistent with our earlier results of measurements published elsewhere (please see: Environ. Monit. Assess (2021) 193:418). In this work, in addition to AAS, we measured the metal concentrations using the technique of Differential Pulse Polarography and found consistent results of the undetected metals. On the other hand, we are in the process to implement the instrument of Inductively Coupled Plasma-Mass Spectrometry. With this instrument, we can obtain limits of detections in the range of ng/L. We are about to conduct our next sampling campaign next November, 2023.

Round 3
Reviewer 1 Report
The authors have addressed my concern. I think this manuscript can be accepted in the current version.